# Egg Freshness Evaluation Using Transmission and Reflection of NIR Spectroscopy Coupled Multivariate Analysis

**DOI:** 10.3390/foods10092176

**Published:** 2021-09-14

**Authors:** Fuyun Wang, Hao Lin, Peiting Xu, Xiakun Bi, Li Sun

**Affiliations:** School of Food and Biological Engineering, Jiangsu University, No. 301 Xuefu Road, Zhenjiang 212013, China; wfy1816491734@163.com (F.W.); xpt201009@163.com (P.X.); bxk360389157@126.com (X.B.); raulsunli@ujs.edu.cn (L.S.)

**Keywords:** near-infrared spectroscopy, non-destructive detection, diffuse transmission spectrum acquisition, freshness

## Abstract

This work presents a novel work for the detection of the freshness of eggs stored at room temperature and refrigerated conditions by the near-infrared (NIR) spectroscopy and multivariate models. The NIR spectroscopy of diffuse transmission and reflection modes was used to compare the quantitative and qualitative investigation of egg freshness. It was found that diffuse transmission is more conducive to the judgment of egg freshness. The linear discriminant analysis model (LDA) for pattern recognition based on the diffuse transmission measurement was employed to analyze egg freshness during storage. NIR diffuse transmission spectroscopy showed great potential for egg storage time discrimination in normal atmospheric conditions. The LDA model discrimination rated up to 91.4% in the prediction set, while only 25.6% of samples were correctly discriminated among eggs in refrigerated storage conditions. Furthermore, NIR spectra, combined with the synergy interval partial least squares (Si-PLS) model, showed excellent ability in egg physical index prediction under normal atmospheric conditions. The root means square error of prediction (RMSEP) values of Haugh unit, yolk index, and weight loss from predictive Si-PLS models were 4.25, 0.031, and 0.005432, respectively.

## 1. Introduction

An egg is an organic vessel that is vulnerable to internal and external environmental factors, causing quality deterioration during storage. The traditional detection methods of egg quality are manual sorting and chemical analysis. There are large errors and high labor intensity in visual observation, and the detection accuracy cannot be guaranteed. The physical and chemical test indexes mainly include Haugh unit, albumen height, pH value, and egg yolk coefficient [1], which evaluate the quality of eggs incubated and sold [2]. Compared with manual sorting, this method has higher accuracy, but this method is destructive and has a great wastage. Neither of the two methods could meet the needs of industrial production. With the development of computer science and non-destructive technology, spectral analysis has become a means of detection. NIR spectroscopy technology has a wide range of research and application in food and pharmaceutical detection, which can realize the detection on the internal and external quality. This method has the advantages of simple operation, low cost, rapid non-destructive testing effect, and mature development at home and abroad.

Except for the temperature and humidity, hens age also brings influence to eggs. The interaction between hens age and storage time impact egg quality significantly. This means that the external and internal factors can affect the quality of eggs, so more attention needs to be paid to the preservation of eggs [3]. A storage experiment was conducted to study the environmental impact on egg freshness. The results indicated that the freshness of eggs could be predicted by the HU regression model, as it reflected the combinations among temperature, humidity, and airflow velocity [4]. Hyperspectral imaging, NIR technology, and 3D laser imaging technology could determine the freshness and measure the internal quality of eggs. Zhang used hyperspectral imaging technology to assess the internal quality of eggs. Six hundred and forty-five white shell eggs were detected by hyperspectral technology and traditional measurement indicators. The hyperspectral images of intact eggs were under the transmittance mode by using a push broom in the spectral range of 380–1010 nm. The successive projections algorithm and support vector regression established a freshness detection model, this model achieved a set of results: a determination coefficient of 0.87, a root mean squared error of 4.01%, and the ratio of prediction to the deviation of 2.80 in the validation set [5]. Coronel-Reyes predicted the storage time for eggs stored at room temperature using NIR spectrometer technology, and a wavelength range between 740 nm and 1070 nm. Six hundred and sixty egg samples were analyzed and developed a robust calibration model. For the storage time of eggs regression, the coefficient of determination (R-squared) was 0.8319 ± 0.0377, and the root mean squared error in the cross-validation test set (RMSECV) was 1.97 days [6]. Giunchi made a qualitative measurement of eggs’ freshness by the FT-NIR reflectance and investigated the relationship between days of storage in the spectral range 833–2500 nm. It was found that 100% of egg samples were correctly discriminated, and the R-squared in the predictive model was up to 0.722, 0.789, and 0.676 for air cell height, thick albumen heights, and Haugh unit, respectively [7]. Zhao employed support vector data description (SVDD) with NIR spectroscopy to deal with the problem caused by the imbalance training samples (where the number of samples needed to be classified varies greatly) in eggs freshness determination. The recognition rate of the SVDD model in the prediction set was 93.3%, and the identification rate of fresh eggs and unfrozen eggs was 93.3% [8]. Kaveh Mollazade developed a new approach to determine the egg’s freshness by 3D laser scanning. It was found that yolk index and Haugh unit can be measured reliably by this method and the root mean squared error (RMSE) reached 0.04 [9].

As introduced above, the traditional manual sorting of egg freshness detection has been gradually replaced by non-destructive detection technology. Acoustic vibration technology, machine vision, and optical detection have been employed in many fields. However, acoustic technology is easy to be disturbed by the noise of the external environment, the reproducibility in the process of machine vision detection is not high, and it is easy to be affected by light and the eggs’ own color. FT-NIR is a non-destructive and pollution-free detection technology with fast imaging technology, spectrum integration, micro area analysis, small sample demand, which in optical detection has the advantages of higher sensitivity, fast measurement speed, and wideband. In this study, NIR spectroscopy combined with multivariate analysis was used to detect the freshness of eggs. The multivariate analysis method could objectively analyze the feasibility of NIR spectroscopy in the detection of egg quality. Traditional models, such as the KNN algorithm, have large amounts of calculations and low prediction accuracy. Support vector machine (SVM), which is sensitive to parameter adjustment and kernel function selection, occupies a large memory. For random forest, the effect of data processing is not obvious. Si-PLS algorithm simplifies non-linear analysis and has obvious prediction results, so Si-PLS was chosen. This study aims to create a new approach with multivariate analysis to predict the storage time of eggs and their physical indexes.

## 2. Materials and Methods

### 2.1. Sample Preparation

A total of 210 brown-shell intact eggs were selected. The selected eggs were from the same hen, and the feeding conditions of the hen met the HACCP Management Technical Specification (NY/T 1338-2007). The egg samples were divided into two groups. One group was stored at 4 °C and the other at 25 °C. Before using NIR to detect the quality of eggs, the eggs stored in the refrigerator were taken out for two hours to make the temperature equal to the ambient temperature, and water droplets were wiped from the eggs’ surface.

### 2.2. NIR Spectroscopy Measurement

For each storage condition (105 eggs selected), the eggs were divided into seven subgroups (each containing 15 eggs). The eggs were stored for 1, 3, 5, 7, 9, 11, and 13 days. After the storage setting time expired, NIR spectra of the eggs from each subgroup were recorded. These spectra were used for the quantitative determination of the relationship between NIR spectra and traditional methods. The diffuse reflectance and diffuse transmittance pattern were used comparatively to obtain the NIR spectra of the eggs. 

#### 2.2.1. Diffuse Reflectance Measurement 

Under the reflectance mode, an Antaris II near-infrared spectrophotometer (Thermo Electron Co., Waltham, MA, USA) with a fiber optic sampling probe was used to record the NIR spectrum. Comparing the whole structure of eggs, it was found that it is easier to obtain the changes of the tissue composition of eggs near the equator of the eggshell than in other parts. Therefore, the light probe is generally placed near the axis of the eggshell equator to illuminate the sample and obtain diffusely scattered light. The experiments were performed in triplicates to reduce the impact of uneven tissue on the results, and the average of these spectral data was used for analysis. Comparing the spectra collected under the conditions of a long wave (1000–2500 nm) and short wave (350–1025 nm), it was found that the reflection result of a long wave is better than that of a short wave, so the selected spectral range was 1000–2500 nm. The data was measured at 3.856 cm^−1^ intervals, which resulted in a total of 1557 variables. Each spectrum was acquired at three different locations under the same light source. The temperature and humidity of the laboratory were controlled at 25 °C and 70%.

#### 2.2.2. Diffuse Transmittance Measurement

The NIR spectrum was obtained in the transmittance mode of the spectrum acquisition system. The system is shown in Figure 1. A spectrometer (MAYA2000+, Ocean Optics Co., Dunedin, FL, USA) was used to obtain visible-NIR transmission data for each egg. The wavelengths of transmittance ranged from 550 to 985 nm with a 0.21 nm increment, which resulted in 2071 variables. The transmittance value of each spectrum was recorded. Three halogen lamps were selected as the light source. The entire eggshell was irradiated evenly as the spectrometer sensor was placed under the egg to obtain the light transmitted through the sample. A whiteboard (thickness = 6.5 mm) made of polytetrafluoroethylene (PTFE) was used as a standard reference spectrum to correct the spectrum before the egg spectrum measurement. Each spectrum was the average of three spectra collected from three different positions under the same light source. Sample, reference, and dark current integration time was 20 ms, and Spectra Suite (Ocean Optics Co., Dunedin, FL, USA) software was used for the spectrometer parameter settings, data acquisition, and storage. The spectrum collected in this experiment was the absorbance value of each wavelength. 

### 2.3. Destructive Measurement 

After collecting NIR spectra for egg storage with 1, 3, 5, 7, 9, 11, and 13 days, the indexes of weight loss rate, yolk index, and Haugh unit of each egg sample were measured.

#### 2.3.1. Weight Loss (%) 

After the temperature of the egg samples reached 20 °C and the humidity was relatively constant, the weight of each egg was recorded on an electronic balance. The accuracy of the electronic balance reaches 1 mg. After collecting the spectrum, the mass of the samples was measured. The weight loss rate of eggs was calculated by using the measured original mass of the sample [10]. The weight loss rate (%) is calculated as follows:$$WR\%=(W_{org}-W_{tst})/W_{org}\times 100\%$$
where $W_{org}$ is the original mass of the fresh egg, $W_{tst}$ is the mass of eggs stored for several days. 

#### 2.3.2. Yolk Index (YI)

Under the condition of keeping the egg yolk intact, the diameter and height of the egg yolk were accurately measured with a vernier caliper (accuracy is 0.1 mm). The ratio of yolk height to diameter is the yolk index [11]. YI was calculated as follows.
YI=h/d
where h is yolk height, and d is yolk diameter.

#### 2.3.3. Haugh Unit (HU)

The average of the three albumens 10 mm away from the yolk was taken as the thick albumen height, and the thick albumen height was then used to obtain the value of HU [12]. HU was calculated as follows:$$HU=100\mathrm{lg}\left(h+7.57-1.7m^{0.37}\right)$$
where *h* is the thick albumen height, *m* is egg mass.

### 2.4. Spectral Data Preprocessing

Data preprocessing could reduce the interference of environmental noise and physical properties of the NIR spectral data. In this experiment, various spectral preprocessing techniques were compared, including standard normal variate transformation (SNV), multiplicative scatter correction (MSC), the first derivative, and the second derivative. These methods could eliminate the phenomenon of baseline drift and enhance small spectral differences and optimize the results. Finally, based on a better prediction, the first derivatives were used as preprocessing techniques in this work. 

### 2.5. Multivariate Calibrations

#### 2.5.1. Principal Component Analysis (PCA) 

PCA is mainly used to simplify data and extract adequate information from the spectral data. NIR spectroscopy data contains an array of multiplication variables overlapping information, so it is necessary to remove irrelevant information from the spectral data. This algorithm could convert the wavelength to a new axis and set the principal component to a new vector. These variables would replace the original data from the multivariate model [13]. 

#### 2.5.2. Linear Discriminant Analysis (LDA) 

In this model, parameters could be divided into observable data and hidden numbers [14]. This model could obtain feature values and make judgments through conditional probability. Linear discrimination could classify the data and then perform dimensionality reduction processing on them.

#### 2.5.3. Synergy Interval Partial Least Squares (Si-PLS) 

Partial least squares (PLS) could analyze the variable matrix Y (the index of egg freshness detection) and the variable matrix X (the spectral data) at the same time [15]. It is often used in NIR spectroscopy to establish regression models to predict the amount of a specific component or the response to the Y-value [16]. To extract valid data, the principal components (PCs) would be obtained from the PCA as the new feature vector of the original spectrum. The results have been evaluated by the full cross-validation of the PLS model to prevent the correction model from overfitting. Because the number of principal components interferes with the effect of PLS, it needs to be optimized [15,17,18]. The best prediction results are based on PLS corresponding to the lowest RMSECV value. 

The principle of Si-PLS is as follows: firstly, it divides the spectral data into many small equidistant regions; secondly, the PLS regression models are established with all possible combinations of two or three sub-intervals, and by each combination of intervals, RMSECV could be calculated. The last step is to choose the lowest RMSECV combined with sub-intervals [19,20].

### 2.6. Software

All algorithms were run on MATLAB V7.0 (MathWorks, Natick, MA, USA) under Windows XP. NIR spectral data were acquired by Result Software (Antaris II System, Thermo Electron Co., Waltham, MA, USA).

## 3. Results and Discussion

### 3.1. Physical Indexes Analysis with Eggs Storage Time Changes

Table 1 shows the physical indexes of eggs freshness stored at temperatures of 4 °C and 25 °C. It could be found that the Haugh unit, yolk index, and weight loss rate have significant changes with the storage time. The rate of weight loss increased significantly as storage time increased every two days. Compared with the eggs stored in the refrigerator, the physical indexes of eggs stored at room temperature showed regular changes, the Haugh unit and yolk index showed a downward trend, and the quality change was relatively stable. It is worth noting that the boundary between grade AA and grade A eggs is a Haugh unit value of 72. Testing eggs remained high fresh (grade AA) within seven (7) days of storage; the freshness began to decline to grade A afterward. The rate of weight loss steadily increased throughout storage. The weight loss rate surged from 0.117% after the first day of storage to 3.286% after 13 days of storage. This represents an increase of nearly 30 times. It can be explained that weight loss is a more sensitive indicator for egg storage time identification.

In the process of testing the egg stored at 4 °C for 13 days, the values of the Haugh unit and yolk index changed irregularly. The change could be attributed to the individual differences among eggs. However, weight loss values still increased continuously but more slowly than those stored at 25 °C. While the relationship between the value of the Haugh unit and the yolk index was minimal when stored at 4 °C and 25 °C, the value of the weight loss rate was significantly related (R^2^ = 0.988) in these two conditions, similar to previously published work. The weight loss rate of preserved eggs increased (*p* < 0.05) as the storage time and temperature increased [21]. These results confirm that weight loss rate is a sensitive index that reflects egg freshness. Table 2 shows the relationship between physical indexes in different storage conditions. There was a significant relationship between the Haugh unit’s values and weight loss rate (R^2^ = 0.975) when the eggs were stored at 25 °C. Meanwhile, there were slight relationships between the other physical indexes stored at 25 °C or 4 °C.

### 3.2. Comparison of Egg NIR Spectra with Different Patterns

The diffuse reflectance and diffuse transmittance patterns were comparatively used to obtain a more effective egg spectra collection pattern. 

Figure 2 shows the comparison spectra of eggs using diffuse reflection (a–c) and transmission modes (d,e). Figure 2a shows that the NIR spectrum obtained from the intact egg is basically overlapped with that obtained from the same egg without contents. In Figure 2b, it is obvious that the spectra almost overlap which is obtained from the eggshell that contained a small amount of albumen and contained a large amount of albumen. Figure 2c shows spectra of the same eggshell with endomembrane and without endomembrane, and the former with remarkably higher values of signal could be observed. It suggests that when the light source passes through the endomembrane of the eggshell, the diffuse reflectance information of egg yolk and protein was blocked by the eggshell, and hardly fed back to the detector. From the experiment above, it can be concluded that, in the diffuse reflectance pattern using commercial NIR spectrometer, only the information of endomembrane of eggshell could be analyzed; it was difficult to obtain information of albumen and yolk inside the eggs.

Figure 2d is the diffuse transmission NIRs of intact egg and eggshell. Figure 2e shows the spectrum of empty eggshell, eggshell with endomembrane, eggshell with a small amount of albumen, and eggshell with a large amount of albumen. It can be found that the spectral data of intact eggs is significantly higher than that of empty eggshells, and the optical signals of egg samples with different contents are also significantly different. This means that the NIR detector can receive the information of light penetrating the eggshell, albumen, and yolk under the condition of diffuse transmission. Therefore, the two detection modes are compared, and the diffuse transmission mode is selected in this paper. 

### 3.3. Data Processed by PCA Algorithm

Figure 3a presents the two-dimensional (2D) space map of eggs laid in room atmospheric storage. PC1 and PC2 explain their differences, respectively. The central tendency of the two-dimensional (2D) space can be seen through the distribution map as shown in the image. Figure 3a shows that the eggs stored at different times have clear differences from each other. Storage 1d, 7d, and 13d could be separated directly in PCA, but the separation effect of the other samples is not apparent, especially the groups of egg samples with 9d, 11d, and 13d that have multiple overlapping parts. Figure 3b shows a two-dimensional (2D) space map of egg samples in refrigerated storage conditions. PC1 interprets 95.8% variances, and PC2 3.07% variances, the cumulative contribution rate of the first two principal components is 98.8%. Compared with eggs in normal atmospheric storage conditions, the cluster trend of the PCA score from egg samples in refrigerated storage conditions is not apparent. To further investigate the classification of egg samples, the LDA algorithm is employed to discriminate egg storage at different times. PCA was used as latent variables in the LDA classifiers. 

### 3.4. Calibration of Models

This study aims to establish a qualitative measurement method for egg storage. This method could also quantitatively predict egg quality in terms of the Haugh unit, yolk index, and weightlessness provided by destructive methods. All egg samples stored at room temperature and refrigerated storage temperature were divided into two subsets. One was called the calibration set used to build a model, and the other was called the prediction set which was used to test the model’s robustness. One spectrum of every three samples was selected in the prediction set to divide the samples into calibration and prediction spectra. Therefore, in each storage condition, the calibration set contained 70 samples, and the prediction set contained 35 samples. Table 3 shows the division of egg samples in the calibration/prediction set, their corresponding Haugh unit, yolk index, and weightlessness values. In qualitative discrimination of egg storage days, the results were assessed by the prediction set’s identification rates. In the quantitative measurement of egg quality, the results were evaluated by the RMSEP and R in the prediction set.

### 3.5. Qualitative Identification of Eggs with Different Storage Time

In this study, LDA was applied to discriminate eggs at room temperature and refrigerated storage temperature on different days. Table 4 shows egg sample discrimination at room temperature with different days using the LDA model. The optimal model could be achieved when the number of PCs was equal to 13. The LDA discrimination model rates were noted with 94.2% and 91.4% accuracy in the prediction set; only a few samples were misidentified. It means that egg samples stored at room temperature on different days could be identified using NIR spectra combined with pattern recognition methods.

The results of egg sample discrimination in refrigerated atmospheric temperature with different days using the LDA model are also shown in Table 4. However, only a minimal number of eggs were correctly identified (54.3% in the training set). This means that most of the egg samples stored at 4 °C for 1 to 13 days could not be discriminated. This is mainly attributed to principal indicators of the freshness of eggs that do not reduce significantly with storage till the 13th day.

### 3.6. Quantitative Analysis of Eggs Freshness Using Si-PLS Models

In this work, Si-PLS models were used to build the relationship between NIR spectra and physical indexes of egg freshness in normal atmospheric and refrigerator storage temperatures. The selection of frequency intervals was first accomplished by Si-PLS. The frequency variables were divided into 25 equidistant subintervals because the utilization of more than this number did not improve the results from previous attempts. 

From Table 5, the NIR spectra combined with Si-PLS models show great ability in eggs physical indexes prediction stored at 25 °C. R in the prediction set in 4 °C is not high enough. That can also be explained by slow changes that occur in the freshness of eggs laid in refrigerated storage conditions, and the characteristic NIR spectra of egg samples are different. Therefore, the NIR spectra method not fits for quantitative analysis of eggs physical indexes in storage condition with low temperature. For egg samples stored at 25 °C, the scatter plot of references measured and NIR predicted by three Si-PLS models in calibration and prediction sets are shown in Figure 4. The changes in protein and water are the main factors that cause changes in the Haugh unit. which is mainly manifested in C-H and O-H groups in NIR spectra. The change of yolk index is mainly caused by the water diffusion of egg white. In the NIR spectra, it is mainly manifested in N-H and O-H groups. The weight loss rate is mainly related to the weight of the egg. The water vapor and CO_2_ in the egg escape through the pores on the eggshell, and the weight of the egg is reduced. In the NIR spectra, the O-H group is mainly present. PLS regression analyses were carried out in the useful spectra range of 666–689 nm and 905–965 nm for Haugh unit; 650–682 nm, 746–777 nm, and 899–957 nm for yolk index; 678–777 nm for weight loss rate, respectively. The spectral regions of 650–777 nm and 899–965 nm were more accurate for measuring egg freshness. Effective spectral region selection by Si-PLS models is shown in Figure 5.

## 4. Conclusions

In this work, the freshness of eggs stored at room and refrigerated temperatures was investigated. The indicators of Haugh unit, yolk index, and weight loss express egg storage time at room temperature conditions. In contrast, only weight loss could excellently express egg storage time in refrigerated storage conditions. NIR spectroscopy appeared to have great potential in quantitative and qualitative analysis of the freshness of shell eggs stored at room temperature. Compared with the diffuse reflectance pattern, egg freshness is more sensitively reflected in the diffuse transmission pattern. NIR spectra, combined with Si-PLS models demonstrated excellent prediction of physical indexes of eggs at room temperature storage.

## Figures and Tables

**Figure 1 foods-10-02176-f001:**
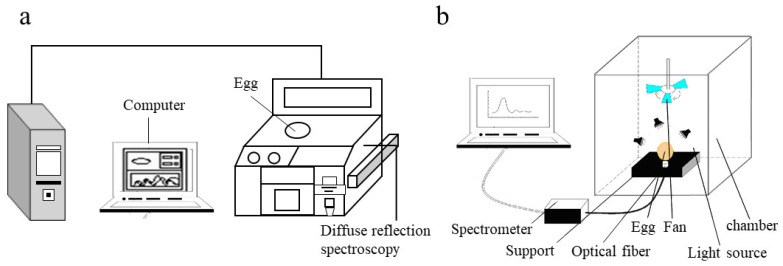
A schematic diagram for NIR spectra collection device using diffuse reflectance mode (**a**) and diffuse transmittance mode (**b**).

**Figure 2 foods-10-02176-f002:**
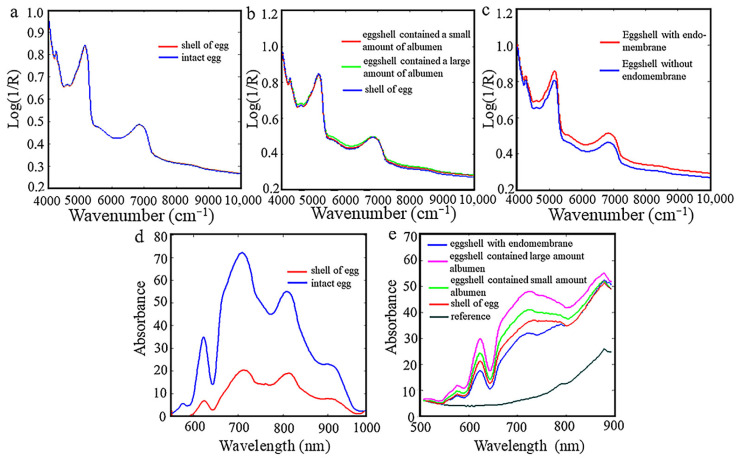
Comparison spectra of eggs using diffuse reflection (**a**–**c**) and transmission modes (**d**,**e**).

**Figure 3 foods-10-02176-f003:**
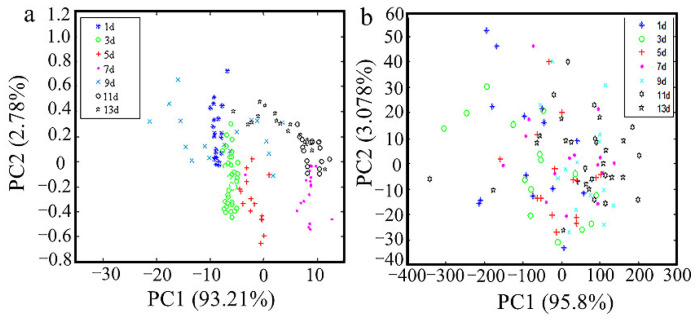
Two-dimensional (2D) space of egg samples represented by PC1 and PC2 stored at temperatures of 4 °C (**a**) and 25 °C (**b**).

**Figure 4 foods-10-02176-f004:**
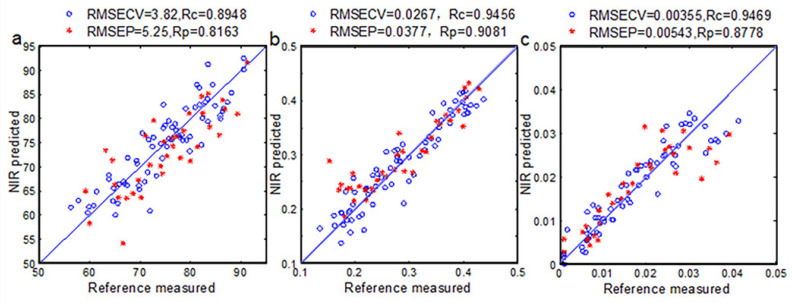
Correlation between reference measured and NIR predicted of the Haugh Unit (**a**), yolk index (**b**) and weight loss rate (**c**) in egg samples. (Storage temperature was 25 °C).

**Figure 5 foods-10-02176-f005:**
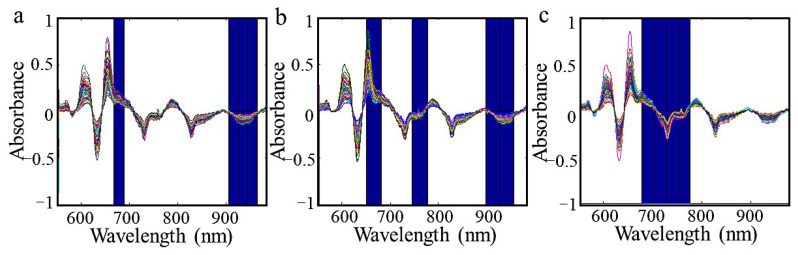
Optimal spectral regions selected by SI-PLS of the Haugh Unit (**a**), yolk index (**b**), weight loss rate (**c**) in egg samples.

**Table 1 foods-10-02176-t001:** Physical indexes of eggs freshness stored at temperature of 4 °C and 25 °C.

Storage Time (d)	Haugh Unit		Yolk Index		Weight Loss (%)	
	25 °C	4 °C	25 °C	4 °C	25 °C	4 °C
1	84.94 ± 4.05 ^a^	85.89 ± 4.73 ^c^	0.36 ± 0.041 ^a^	0.42 ± 0.027 ^b^	0.117 ± 0.0246 ^g^	0.173 ± 0.043 ^g^
3	81.75 ± 4.59 ^ab^	89.78 ± 2.65 ^b^	0.32 ± 0.11 ^ab^	0.443 ± 0.027 ^a^	0.637 ± 0.055 ^f^	0.325 ± 0.066 ^f^
5	80.01 ± 1.85 ^ab^	92.75 ± 4.17 ^a^	0.32 ± 0.042 ^ab^	0.433 ± 0.025 ^ab^	0.954 ± 0.132 ^e^	0.469 ± 0.054 ^e^
7	79.35 ± 4.09 ^b^	92.63 ± 2.76 ^a^	0.28 ± 0.038 ^bc^	0.443 ± 0.02 ^a^	1.526 ± 0.281 ^d^	0.66 ± 0.18 ^d^
9	70.95 ± 5.11 ^c^	94.57 ± 3.25 ^a^	0.26 ± 0.11 ^c^	0.437 ± 0.034 ^ab^	2.068 ± 0.299 ^c^	1.04 ± 0.14 ^c^
11	65.96 ± 5.62 ^d^	94.65 ± 2.98 ^a^	0.24 ± 0.032 ^cd^	0.449 ± 0.02 ^a^	2.760 ± 0.446 ^b^	1.22 ± 0.17 ^b^
13	59.19 ± 1.87 ^e^	92.05 ± 2.56 ^ab^	0.20 ± 0.031 ^d^	0.430 ± 0.019 ^ab^	3.286 ± 0.467 ^a^	1.52 ± 0.20 ^a^
R^2^	0.92	0.57	0.97	0.13	0.99	0.97

Comparison of significance in the same column; significance level α = 0.05.

**Table 2 foods-10-02176-t002:** Relationship of physical indexes (R^2^) of eggs stored at temperature of 4 °C and 25 °C.

	Haugh Unit		Yolk Index	
	25 °C	4 °C	25 °C	4 °C
Haugh Unit	--	--	--	--
Yolk index	0.3	0.498	--	--
Weight loss (%)	0.975	0.464	0.307	0.0943

**Table 3 foods-10-02176-t003:** Statistics of egg indexes in calibration and prediction samples sets.

Indexes	Sample Sets	Group A (25 °C)		Group B (4 °C)	
		Range	Mean ± SD	Range	Mean ± SD
Haugh Unit	Calibration	56.33–90.55	74.69 ± 8.44	80.24–101.03	90.48 ± 4.37
	Prediction	59.19–91.25	75.23 ± 8.36	76.26–100.45	92.32 ± 4.32
Yolk Index	Calibration	0.13–0.43	0.28 ± 0.08	0.38–0.48	0.43 ± 0.028
	Prediction	0.15–0.42	0.28 ± 0.08	0.35–0.49	0.43 ± 0.023
Weight Loss Rate	Calibration	0.00091–0.004	0.016 ± 0.011	0.001–0.02	0.0079 ± 0.049
	Prediction	0.00095–0.039	0.016 ± 0.011	0.0009–0.015	0.0075 ± 0.047

**Table 4 foods-10-02176-t004:** Results of egg storage time discrimination in different storage conditions using the LDA model.

Storage Time (d)	Group A (25 °C)		Group B (4 °C)	
	Train	Test	Train	Test
1	9/10	4/5	3/10	3/5
3	10/10	5/5	6/10	1/5
5	10/10	5/5	4/10	1/5
7	9/10	4/5	9/10	1/5
9	9/10	5/5	7/10	0/5
11	9/10	5/5	5/10	0/5
13	10/10	5/5	4/10	3/5
Total (%)	94.2	94.2	54.3	25.7

Train: The subset used to train the model; Test: Test a subset of the trained model.

**Table 5 foods-10-02176-t005:** Performance of optimal Si-PLS models for egg physics indexes prediction.

Components	Group A (25 °C)				Group B (4 °C)			
	Effective Spectral Region/nm	PCs	RMSEP	R(t)	Effective Spectral Region/nm	PCs	RMSEP	R(t)
Haugh Unit	666–689, 905–965	10	4.25	0.816	622–644, 844–866, 910–954	3	5.23	0.436
Yolk Index	650–682, 746–777 899–957	8	0.031	0.908	704–729, 754–778, 828–853, 877–904	8	0.0358	0.174
Weight Loss Rate	678–777	7	0.00543	0.877	666–716, 755–777, 844–868	7	0.0127	0.459

R(t) means the correlation coefficients in prediction set.

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
