# Peer review of "Egg Freshness Evaluation Using Transmission and Reflection of NIR Spectroscopy Coupled Multivariate Analysis"

_foods, 2021, doi:10.3390/foods10092176_

Round 1
Reviewer 1 Report
In this paper, the authors report that the diffuse transmission NIR-spectroscopy with the Si-PLS model can be a promising modality for nondestructively evaluating the freshness of eggs. For analyzing the physical quality indexes corresponding to their freshness, Haugh unit, yolk index, and weight loss were investigated in two different storage temperatures (25°C, 4°C).
However, I found some previous similar reports on the egg freshness evaluation using the transmission NIR-spectroscopy combined with
PLS model, even though they didn't use the synergy interval PLS (Si-PLS) model. Further, suppose the authors want to focus on the Si-PLS model being a prominent and new approach compared to the existing research. In that case, it is necessary to show the results what is superior to the conventional method. In the submitted study, I can't see significant improvements in the prediction performance except employing the Si-PLS model to predict the freshness of the eggs.
Hence, this work in its current form is not sufficient enough to be published in Foods. Apart from that, I have some minor comments as well:
1. I found some typos such as,
-P3 L45: Hugh unit, album height-> Haugh, albumen
-P3 L65, L70: before nm, blank spaces are required.
-P5 L126: before ms, blank spaces are required.
-P4 L81: water vapor, CO2 -> '2 ' is a subscript character.
-P5 L107: internal issue -> tissue
-P5 L116: reflectance->transmittance
-P7 L149: 10 nm -> 10 mm
-P9 L219 R2 -> '2 ' is a superscript character.
-P14 L317 period-> comma
2.In the introduction part, it is necessary to explain sufficient background (like drawbacks and benefits on conventional tools) and justification for the new multivariate analysis approach. Comparison of temperature conditions or measurement conditions does not seem to have much significance.
3. It would be nice to see a schematic diagram for diffuse reflection spectroscopy in section 2.2.1, even if it is a commercial spectrometer.
4. Please indicate the measurement precision of the electronic balance in P6 L137.
5. Please indicate the SW for the reflectance and transmittance spectra separately.
6. Please denote the meaning of index a,b,c,d,e in table 1
7. Please add the related references in P9 L212.
8. In figure 2,
a-c, Log1/R -> Log(1/R)
d-e, wavenumber -> wavelength(nm)
9. In table 3-5, It would be better to move the description of the temperature below to beside Group numbers.
10. What is R(t) in table 5?
11. For clarifying the results, please denote Group A or temperature in the figure 4 caption.
Reviewer 2 Report
The manuscript entitled: "Egg Freshness Evaluation Using Transmission and Reflection of NIR Spectroscopy Coupled Multivariate Analysis" by Wang et al., describes the application of NIR spectroscopy followed by multivariate statistical analysis on the evaluation of egg freshness. The manuscript is well designed and the authors have properly described the experimental workflow regarding the instrumental/statistical analyses.
1) The standard of English must be improved throughout the manuscript. Several errors and/or misleading sentences have been detected.
2) In the Abstract section, authors are invited to deliver a clear take-home message, putting more emphasis on the novelty of their application as well.
3) Please, try to differentiate the keywords from those used in the Title section.
4) According to a check on different databases (such as Scopus or Science Direct), I found very similar works already published 10 years ago. For example, Lin et al. (2011) evaluated the freshness of eggs using NIR spectroscopy and multivariate data analysis. More recently, Dong et al. (2019) demonstrated how to keep the prediction ability of egg freshness models on a new variety based on VIS-NIR spectroscopy methods. Therefore, the authors need to clearly differentiate their work from the already existing methods by making realistic comparisons and stating the advantages and limitations of the FT-NIR approach.
5) The experimental plan is well designed, starting from the sample collection and preparation, up to the multivariate statistics. Have the authors tried the application of OPLS-DA supervised modeling?
Round 2
Reviewer 1 Report
I can't recheck the Figure 2, but believe that the author fixed it in the revised manuscript. Before publishing, please recheck thoroughly the revised version.
Reviewer 2 Report
The authors answered in a proper way to each major concern. The manuscript can be accepted in the present form.